# Canoe polo Athletes’ Anthropometric, Physical, Nutritional, and Functional Characteristics and Performance in a Rowing Task: Cross-Sectional Study

**DOI:** 10.3390/ijerph192013518

**Published:** 2022-10-19

**Authors:** Elena Marques-Sule, Anna Arnal-Gómez, Lucas Monzani, Pallav Deka, Jairo P. López-Bueno, Manuel Saavedra-Hernández, Luis Suso-Martí, Gemma V. Espí-López

**Affiliations:** 1Department of Physiotherapy, University of Valencia, C/Gascó Oliag, 5, 46010 Valencia, Spain; 2Physiotherapy in Motion, Multispecialty Research Group (PTinMOTION), C/Gascó Oliag, 5, 46010 Valencia, Spain; 3Ivey Business School, Western University, 1255 Western Rd, London, ON N6G 0N1, Canada; 4College of Nursing, Michigan State University, East Lansing, MI 48824, USA; 5Department of Physical Therapy, University of Almeria, Carretera Sacramento s/n, 04120 Almería, Spain; 6Exercise Intervention for Health (EXINH), C/Gascó Oliag, 5, 46010 Valencia, Spain

**Keywords:** canoe polo, physical attributes, motivation, nutritional sciences, performance predictors

## Abstract

Understanding the physical, functional, mental, and nutritional attributes of canoe polo athletes is essential for training and development. Forty-three canoe polo athletes (mean age: 21.54 ± 6.03) participated in the study and were assessed for: anthropometric measurements, exercise motivation, eating habits, adherence to the Mediterranean Diet, and physical and functional abilities. Correlation and multivariate analysis were conducted. Individual performance in a rowing task showed body mass index (β = 0.41) and female gender (β = 0.34) to be the strongest anthropometric predictors, whereas body fat (β = −0.35) and triceps brachii skinfold fatty tissue (β = −0.35) were the strongest negative predictors. Pushing strength (β = 0.37) and range of motion with internal rotation (β = 0.30) were the strongest physical predictors. The physical dimension of the Exercise Motivation Index was a significant psychosocial predictor (β = 0.27). Senior participants had a higher waist–hip ratio (*p* = 0.04, d = 0.66), arm circumference (*p* = 0.03, d = 0.68), handgrip strength (*p* < 0.01, d = 1.27), and push strength (*p* < 0.01, d = 1.42) than under 21-year-olds. Understanding the highlighted sport-specific characteristics of canoe polo athletes can help trainers to design programs at all levels to optimize performance.

## 1. Introduction

Canoe polo is a dynamic team sport with growing popularity in recent years. This sport was first played in England and the number of people practicing it has gradually increased worldwide. Currently, there is an under 21 years old (U21) competition category that is growing in popularity due to the increasing success of the sport [1].

One of the most differential characteristics of this sport is that players are required to develop a great number of physical capacities such as strength, speed, or even endurance for executing common movements such as sprinting, paddling, or dribbling. Furthermore, canoe polo requires high-intensity bursts of sprinting, interspersed with short periods of low-to-moderate intensity paddling, which makes it a very specific physically demanding sport [2,3]. Moreover, it involves overhead throwing, which generates a series of risks on the shoulder joint complex and alterations in range of motion (ROM) [4].

The increasing popularity of this sport and of the players who practice it has resulted in the need for a greater scientific approach to canoe polo. However, very few studies have focused on canoe polo players. Alves et al., while examining the anthropometric, physiological, and performance characteristics of international canoe polo athletes, found higher upper-body anaerobic power compared to other sports [2]. Forbes et al. conducted a study which aimed to evaluate the characteristics of high-performance canoe players in order to determine the main physical demands of this sport [3]. The findings of these studies reported that athletes with high levels of upper body aerobic and anaerobic fitness tend to perform better in this sport [2,3]. The rest of the publications related to the canoe polo athletes are mainly focused on increasing sports performance and assessing the effect of different training programs [5,6,7].

Previous studies have highlighted the importance of anthropometric and body composition characteristics for high-demanding team sports players; however, among the 21 analyzed sports, canoe polo was not included [8]. Moreover, nutritional habits have been shown to significantly influence the adequate sports performance of athletes [9].

On the other hand, as a previous review has stated, the best assessment for a patient, whether or not they are an athlete, is one where all the functions of the musculoskeletal system are analyzed from their different perspectives [10]. Thus, physical performance assessments and functional tests have recently gained interest since they have been shown to be a good post-surgical marker of recovery prognosis, and they also help to determine the risk of falls and to define the range of performance improvement or risk of injury in a predictive way [10]. Moreover, some of them, such as the 1 min sit-to-stand test or the 1RM test, have demonstrated their validity when assessing strength [11].

Therefore, in addition to understanding the physical demands of any sport, for optimizing performance, a good understanding of the functional, mental, and nutritional attributes is equally essential. Analyzing the physical characteristics of canoe polo athletes along with these attributes is necessary for designing better training programs, avoiding injuries, and improving performance. Thus, the objectives of this study were to analyze the anthropometric, nutritional, physical, functional, and psychosocial characteristics of canoe polo athletes and to explore the relative contribution of these variables in predicting performance in a rowing-specific strength exercise (as a proxy of individual performance in canoe polo). Additionally, we evaluated the differences in these variables between senior and U21 athletes.

## 2. Materials and Methods

### 2.1. Participants

Of the 50 eligible professional canoe polo athletes, 43 agreed to participate and were included in the study (Figure 1). Athletes were recruited from a professional team in Valencia (Spain) and assessed between January and April 2019. Inclusion criterion were being professional canoe polo players and being willing to participate. Exclusion criteria were presence of pain (including chest pain) at the time of the study; injured players without current professional activity; and the presence of vestibular, ocular, or postural deficiencies.

### 2.2. Study Design

A cross-sectional study design with a non-probabilistic sample was carried out at the laboratories of the University of Valencia to analyze anthropometric, nutritional, physical, functional, and psychosocial variables of canoe polo athletes. The design followed the international recommendations of Strengthening the Reporting of Observational Studies in Epidemiology [12]. All participants received an explanation of the study procedures and provided written informed consent before their inclusion. Procedures were planned according to the ethical standards of the Declaration of Helsinki and approved by the Ethics Committee of the University of Valencia (H1509818565591).

### 2.3. Measurements

Demographic variables were evaluated through a questionnaire where participants were asked about their age and gender, as well as sport-related variables such as years competing in canoe polo, number of training hours per week, and prior rowing injuries. For the latter, participants were asked to indicate whether they suffered (a) back or (b) prior shoulder injuries (0 = “NO”; 1 = “YES”). Based on prior studies [13], we anticipated that prior rowing injuries, particularly upper limb injuries, might reduce our participants’ performance in a rowing activity.

#### 2.3.1. Anthropometric Variables

The following anthropometric measurements were conducted three times and the average was used for analysis:

Body fat and body mass index: An acoustic bioimpedance scale, model Tanita BF350 Body Composition Analyser, was used. Participants were asked to step on and position the soles of their feet in contact with the four sensors of the scale. Through acoustic bioimpedance, weight, body mass index (BMI), and body fat (%) were obtained once the height, age, and gender had been entered. This tool has been shown to have adequate reliability for assessing weight and body fat [14].

Body perimeters: A flexible SECA Model 201 tape measure with an accuracy of 1 mm, specially designed for measuring body perimeters, was used to measure: waist and hip perimeter, waist–hip ratio, and arm and thigh circumference following standards previously described [15]. Waist perimeter was classified as low, high, and very high following clinical guidelines [16].

Skinfold: A Quirumed 842-SK-001 calliper with an accuracy of 2 mm was used for the measurement of the triceps brachii skinfold fatty tissue. The measurement was conducted three times on the dominant arm at the level of the mid-point between the acromion process and the elbow joint, on the mid-line of the posterior surface of the arm (over the triceps muscle). The average was selected for analysis [17].

#### 2.3.2. Nutritional Variables

Healthy eating: For the Healthy Eating Index-Spanish (HEI-S) [18] questionnaire, participants provided answers about the consumption of certain foods: cereals and their derivatives; vegetables; fruits; milk and its derivatives; meat; pulses; sausages and cold cuts; sweets; sugared soft drinks; and variety of diet. These 10 variables are divided into 5 frequency of consumption categories: daily intake; three or more times a week but not daily; once or twice a week; less than once a week; and never or hardly ever. Each variable was scored from 0 to 10, with 10 meaning fulfilment of the recommendations of the Spanish Society for Community Nutrition. Then, the HEI-S was calculated by adding up the score of each of the 10 variables; thus, 100 points was the theoretical maximum. The total score was classified in three categories: >80 points, “healthy eating”; >50 points, “needs changes”; and 50 points, “unhealthy eating”.

Adherence to the Mediterranean Diet: The Mediterranean Diet adherence screener (MEDAS-14) consisted of 14 questions related to the Mediterranean diet, validated for the Spanish population. Participants’ responses to each question were valued as 0 or 1; thus, the total adherence score varied between 0 (minimum) and 14 points (maximum), and then it was stratified by 3 categories of adherence to the Mediterranean Diet: low (≤5 points), moderate (6–9 points), or high (≥10 points) [19].

#### 2.3.3. Physical Variables

Shoulder rotation range of motion: To obtain the active ROM in shoulder rotations a conventional analogue polymer goniometer, Saehan model Grip, was used with an accuracy of 2 degrees. External rotation (ER) and internal rotation (IR) of the dominant upper limb were evaluated. The measurement protocol was implemented as previously described and shown to be reliable [20].

Handgrip strength: This was assessed using a Jamar hydraulic hand dynamometer model SP-5030J1, with the standard testing position and instructions for handgrip measurement (kg) as previously recommended [21]. Participants were instructed to sit upright, testing arm at side, elbow flexed at 90°, and forearm in neutral position, then had to squeeze the dynamometer handle once as hard as possible; each hand was measured 3 times consecutively with 10 s pauses between each and the maximal was recorded for analysis [22]. The measurements with the Jamar dynamometer present excellent test–retest reliability for preferred and non-preferred hands (intraclass correlation coefficient (ICC) = 0.81–0.99 for men; ICC = 0.83–1.0 for women) and excellent intra-rater (ICC = 0.94 and 0.98) and inter-rater reliability (ICC = 0.98) [23].

Push strength—overall: To evaluate the upper limb thrust force, an estimate of the maximum load (kg) the participant is capable of moving vertically once on a flat bench press was made, i.e., the 1RM was evaluated in the bench press exercise. 1RM refers to the maximum load (kg) a person can bear in a given movement, and it defines the maximum force for a particular exercise. The estimation of this variable was made using the validated iPhone application “PowerLift” (version 2.8) [24]. This application, by means of the instruction of 3 variables (movement ROM (cm), load (kg), and load moving speed (m/s)), reliably estimates the 1RM (kg). Four measurements were made with a 2 min and 30 s pause between each, and the load was progressively increased with each attempt (approximately 60–80% of the athlete’s theoretical load). This smartphone application has been validated for research purposes [24].

Push strength—rowing specific: We used a similar approach to measure our participants’ ability to lift weight in a rowing machine (Seal Row/Chinese Row Bench), as a proxy for individual rowing performance. We anticipated that higher weight lifting ability in a rowing machine would likely predict posterior performance when rowing [25].

#### 2.3.4. Functional Variables

Countermovement jump (CMJ) test: CMJ was used to evaluate lower limb power. Participants started from an upright position with hands on the iliac crests, and then were asked to perform a lower limb flexion followed quickly by an extension which triggered the jump without helping themselves with the upper limbs to apply inertia. This assessment was performed using the validated iPhone application “My Jump 2” (1.0.6 version) (Apple Inc., Cupertino, CA, USA) [26]. Firstly, a profile was created in the application for each participant where the following information was entered: body weight, lower limb length (from trochanter to toe tips with maximum plantar flexion), and height (from the ground to trochanter with lower limbs in semi-stance, both knees at 90° flexion). Participants were then asked to perform the CMJ, and the smartphone, by using slow motion, calculated the exact height (cm) of the jump [26].

Push-up test: Upper limb functional capacity was established by the 1 min push-up test. Push-ups were performed with 4 supports on the floor: both feet and both hands. From this plate position, flexion-extension of the upper limbs was performed as many times as possible for one minute. The higher the number of repetitions, the greater the upper limb functional capacity [27].

Sit-to-stand test: Lower limb functional capacity was established by the 1 min sit-to-stand test. This assessment consisted of getting up from and sitting down on a chair (45 cm high) as many times as possible for one minute. Two series were performed, and the average of both measurements was obtained. The higher the number of repetitions, the greater the lower limb functional capacity [28].

#### 2.3.5. Psychosocial Variables

Motivation to exercise: This was assessed using the Exercise Motivation Index questionnaire, which consists of 15 statements followed by a five-point Likert-type rating scale for each statement, ranging from “0 = Not important” to “4 = Extremely important” [29,30]. This scale has three dimensions: (a) physical, (b) psychological, and (c) social motivation. Cronbach’s alpha for this sample was α = 0.76 for the physical motivation dimension; α = 0.70 for the psychological dimension; and a borderline acceptable α = 0.61 for the social dimension.

### 2.4. Statistical Analysis

The Statistical Package for Social Sciences (SPSS 22, SPSS Inc., Chicago, IL, USA) was used for all analyses. The level of significance for all tests was *p* < 0.05. In the data analysis, descriptive statistics were used to show the data on the continuous variables, presented as mean and standard deviation (SD), 95% confidence interval (95% CI), and frequency (%).

Mean differences: We conducted a χ^2^ test to compare the differences between the categorical variables (nominal). The Shapiro–Wilk test was used for the normality tests. We applied the Student’s *t*-test for independent samples as a statistical test to compare the continuous variables between groups. We calculated the effect size (Cohen’s d) to compare the study variables.

Correlational analyses: Following Cohen’s [31] recommendations, we classified effects into small (0.20 to 0.49), medium (0.50 to 0.79), or large (>0.8). The relationships between physical and functional variables with nutritional and anthropometric measures were examined using Pearson’s correlation coefficients. A Pearson’s correlation coefficient greater than 0.60 indicated a strong correlation, a coefficient between 0.30 and 0.60 indicated a moderate correlation, and a coefficient below 0.30 indicated a low or very low correlation.

Multivariate analyses: We conducted multivariate regression analyses to predict variance in rowing strokes. To this end, we transformed all predictors into z-scores to (a) allow comparison between the relative effect of each predictor and (b) to reduce potential collinearity issues in our linear model. Thus, the resulting standardized correlation coefficients depict the relative effect size of each predictor on our outcome criterion, ranging from β = −1 to 1. After conducting preliminary analyses, we excluded any variable (height, weight; hip and waist perimeter) that was used in the creation of ratio-type variables (BMI; hip–waist ratio, respectively), as they presented unacceptable variance inflation scores and standardized coefficients larger than 1.

Finally, we report Cohen’s f^2^ for the overall final model (which is the regression equivalent of Cohen’s d), as well as the statistical power achieved (1 − β). With an alpha value of 0.05, a value of 1 − β above 0.95 would ensure that the likelihood of making a Type II error is also below 0.05.

## 3. Results

All the demographic, anthropometric, nutritional, physical, functional, and psychosocial variables are depicted in Table 1.

### 3.1. Mean Differences Analyses

Participants were divided according to their age category into senior (n = 20) and U21 athlete groups (n = 23), with a mean age of 26.9 ± 4.62 and 16.87 ± 1.48 years old, respectively. The senior group had a mean of 9.25 ± 5.69 years of experience in the sport, while the U21 group had 2.74 ± 1.66 years. The hours of practice per week in each group were similar, being 9.55 ± 3.76 for the senior group and 8.50 ± 2.91 for the U21. Table 2 shows differences between the groups in anthropometric, physical, functional, and psychosocial variables. Firstly, the senior group showed significantly higher scores in the waist–hip ratio (*p* = 0.04, d = 0.66) as well as in arm circumference (*p* = 0.03, d = 0.68). In relation to physical variables, the senior group showed higher handgrip strength and a greater push strength than the U21 (*p* < 0.01, d = 1.27; *p* <0.01, d = 1.42, respectively). Finally, higher values were also found in lower limb (*p* = 0.02, d = 1.26) and upper limb (*p* < 0.01, d = 0.74) functional capacity for the senior group.

### 3.2. Correlation Analyses

Figure 2 shows that hand grip strength had the strongest correlations with anthropometric variables, with the exception of tricipital skinfold. These correlations were low and moderate except with arm circumference (r = −0.71; *p* < 0.01). Similar correlations were found between push strength and anthropometric variables, except for hip perimeter and tricipital skinfold. Regarding functional variables, a correlation between CMJ and arm circumference was found (r = −0.51; *p* < 0.01) as well as with thigh circumference and tricipital skinfold (r = 0.33 and −0.46; *p* < 0.05). No correlations were found for sit-to-stand tests or nutritional variables.

### 3.3. Multivariate Analyses

Table 3 shows the results of our regression model. Our model presented an excellent fit to the data, explaining 93.6% of the variance in 1RM rowing (F(2,21) = 5.31, *p* < 0.01; adjusted R^2^ = 0.87). More precisely, BMI (β = 0.41 ***) and anatomical sex (β = 0.34 ***) were the strongest anthropometric predictors of rowing strength. One counter-intuitive finding was that, in our model, female participants showed a higher score in the rowing exercise than their male counterparts. Instead, variables related to fatty tissue, such as overall levels of body fat (β = −0.35 ***) and triceps brachii skinfold fatty tissue (β = −0.35 ***), were the strongest negative predictors of 1RM rowing strength.

Not surprisingly, the overall pushing strength (β = 0.37 ***) and ROM IR (β = 0.3 ***) were the strongest physical predictors of 1RM rowing, given that canoe polo requires frequent movements in the upper torso. Finally, of our three psychosocial predictors, only the physical dimension of the Exercise Motivation Index was a significant predictor of 1RM rowing (β = 0.27 ***). A second unpredicted yet plausible finding was that prior shoulder injuries (β = −0.24 ***) reduced the 1RM rowing test score.

## 4. Discussion

The present study highlights the anthropometric, nutritional, physical, functional, and psychosocial characteristics of a sample of professional canoe polo players. Globally, anthropometric variables follow normality with a trend of increased scores in body fat. Nutritional characteristics indicated these athletes needed changes in their diet and stronger adherence to the Mediterranean diet. Healthier trends in physical and functional characteristics were observed, although with relatively low parameters in certain variables such as ROM. Moreover, these characteristics when analyzed by age showed some statistical differences between both groups.

Anthropometric measurements of the BMI of participants were “healthy weight”, but with a mean score near to being “overweight” [17]. In this sense, it has to be taken into consideration that compared with the general population, the influence of large muscle mass on BMI in athletes may misclassify them as overweight [32]. Thus, BMI must be interpreted with caution in these highly muscular athletes since it may be a less accurate measure of adiposity for them [17]. It may be useful to consider parameters such as waist perimeter and/or body fat. Most participants had a low score on the waist parameter. As for body fat, our results showed relatively high values compared to previous studies of canoe polo athletes [2] and of similar sports [33], which may be because our participants, despite being professionals, are not elite athletes, as in the abovementioned studies, and consequently have different intensity and frequency of training. Moreover, we found no significant differences in body fat or in training practice hours between senior and U21 athletes. As mentioned in Alves et al., technical and tactical factors may have an influence on body composition in elite canoe polo athletes [2], although future research may help determine this relationship. Canoe players should be educated on following guidelines on weight management and be provided general advice on a healthy weight and lifestyle [17].

To our knowledge, this is the first study of canoe polo players which has measured body perimeters, which have been described as a health indicator in adults, especially the waist–hip ratio. Both senior and U21 athletes had waist–hip ratio results within healthy parameters [34], but there were significant differences between both groups, as well in arm circumference, likely due to the presence of a higher muscle mass in the senior group. As in previous studies, the gain in muscle mass in senior paddlers can be observed in their arms and legs [35]. The maturity and growth status of athletes seem to influence arm muscle parameters, which may be a necessary attribute for this sport. Thus, when conditioning and training canoe polo athletes, these factors and body build parameters may be a helpful reference. Nutritional habits, as previously stated, including a correct nutritional plan and a healthy diet, are intricately related to sports performance. Moreover, nutrition is related to anthropometric parameters, and, in turn, body mass and composition are important for aquatic sports athletes [36]. However, our results showed overall a low profile regarding nutritional habits. Therefore, nutritional support may be taken into consideration when training these athletes.

Regarding physical variables, it should be highlighted that the shoulder is one of the most important joints in this sport since it is used for the transmission of force and overhead throwing. This may result in an overuse of this joint as in throwing sports [37]. Our results showed a relatively low ROM in comparison to water-polo, a throwing sport [4]. This factor has been considered a risk factor for shoulder injuries [4], and thus it should be considered when conditioning canoe polo athletes, especially senior players, since they showed lower values in our study. Handgrip strength has been gaining functional importance for the measure of sport-specific movements, especially in water sports since the hand makes contact with an implement and/or object [38]. In canoeing, due to the use of rowing, the hand grip is the last bodily point of contact within the kinetic chain propelling oneself through the water, and it is also one of the most important measures in throwing sports. In our study, significantly higher values were found in the group of senior athletes, and hand grip showed a correlation with most of the anthropometric variables. Taking into account that some research has associated grip strength with increased sports performance [39], this variable is one to be assessed and considered for these athletes, especially the U21 group.

Regarding functional tests, these provide an important measure of performance and injury prevention in athletes and are widely used in rehabilitation processes to mark the return to sporting activity. In relation to countermovement jump, it should be noted that the majority of the existing literature defines the use of the My Jump 2 iPhone application as valid and reliable, without significant differences compared to the evaluations with force platforms [26,40]. On the other hand, the American College of Sports Medicine and the National Strength and Conditioning Association (NSCA) recommend the use of the push-up test without setting a specific time for the test [41,42]. Moreover, the ACSM recognizes that any exercise that respects the principles of fatigue and repetitive muscle activity is likely to serve as an assessment, thus highlighting the principle of individualization in clinical evaluation and follow-up [41]. The 30 s sit-to-stand test has been catalogued as a way to assess functional capacity in older adults [43] However, if we look at the evaluation methodology proposed by the ACSM, both the 1 min sit-to-stand test and the 1 min push-up test are evaluations that fit the definition of muscular resistance or endurance, since the ability of a muscle group to perform repetitive actions for a period of time sufficient to cause local fatigue is assessed [41] Overall, our data can provide a reference for trainers and health professionals involved in the rehabilitation of canoe polo athletes. Despite this, it should be noted that there is great heterogeneity among the tests and more research is needed on this topic [38]. Our results are above the reference values obtained in non-sporting populations but show differences with other sports such as volleyball or basketball [44]. As some studies have previously suggested, each sport may lead to certain adaptations that may influence the performance of these tests [45].

On the other hand, the psychosocial variables (motivation) were a significant predictor of performance. Our results show that participants’ concern with their physical health and body image predicted their performance in the 1RM rowing test. This means that participants were more interested in rowing to ensure their physical health than rowing just to “keep up” with peers or significant others. This dimension confirms the importance of motivation in this sport, with aspects such as wanting to have physical endurance, the importance of physical well-being, good physical health, and a good body image. Therefore, motivation is an important aspect for trainers or physiotherapists who work with canoe polo players. Moreover, in future studies, motivation variables could be included to improve performance.

Some limitations need to be considered in the interpretation of our results. Firstly, the sample size is limited, and the selected athletes, although professionals, may not be elite as in other similar studies. Secondly, some procedures are new measurement tools, which may be a limitation in comparison with other similar studies. Moreover, it would have been interesting to include movement analyses that could provide more data about this sport. On the other hand, the fact that prior shoulder injury remains a significant predictor in the regression suggests that in future studies it should be controlled, at least statistically, by entering it as a covariate or dividing it into groups with or without prior shoulder injury. As strengths, this study offers evidence regarding the characteristics of canoe polo athletes and highlights the need to increase knowledge of these types of variables, which could help improve the performance of canoe polo players or help prevent injuries.

## 5. Conclusions

This study provides relevant information about the anthropometric, physical, functional, and psychosocial characteristics of professional canoe polo players. Due to the scarce number of studies about this discipline and the growing popularity of this sport, these variables of clinical interest can serve trainers and health professionals to develop better training and rehabilitation programs for canoe polo athletes. Additionally, the differences found between senior players and U21 may be useful and serve as a reference for developing training programs.

## Figures and Tables

**Figure 1 ijerph-19-13518-f001:**
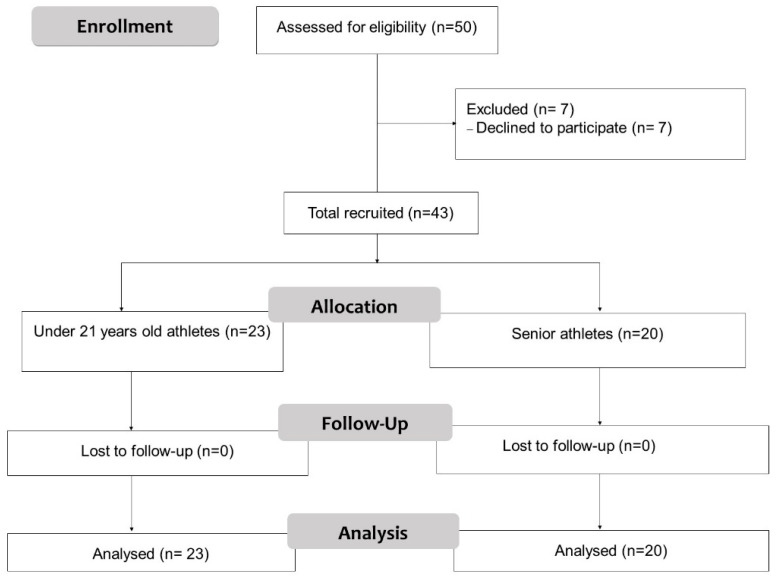
Flow diagram of study participation.

**Figure 2 ijerph-19-13518-f002:**
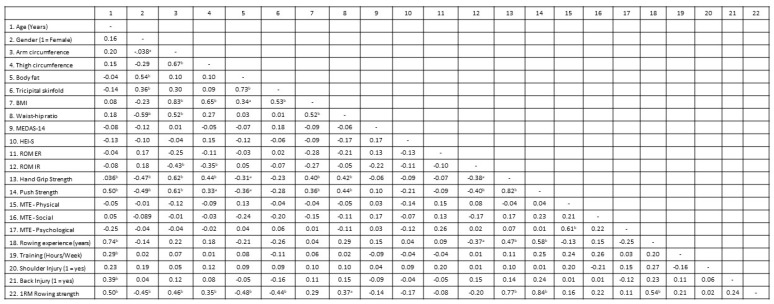
Multivariate analysis—correlation matrix (N = 43). Note: ^a^
*p* < 0.05; ^b^
*p* < 0.01; ROM: range of motion; ER: external rotation; IR: internal rotation; CMJ: countermovement jump; BMI: body mass index; HEI-S: Healthy Eating Index-Spanish; MEDAS-14: Tool of Adherence to the Mediterranean Diet; MTE: motivation to exercise.

**Table 1 ijerph-19-13518-t001:** Results of demographic, anthropometric, physical, functional, and psychosocial variables.

Variable	Total (n = 43)
Demographic variables	
Age (years), mean ± SD	21.54 ± 6.03
Gender (men/woman), n (%)	36 (83.7)/5 (16.3)
Exercise Motivation	
Exercise Motivation Index—Physical	1.79 ± 0.51
Exercise Motivation Index—Social	1.33 ± 0.63
Exercise Motivation Index—Psychological	1.53 ± 0.51
Anthropometric variables	
BMI (kg/m^2^), mean ± SD	23.27 ± 2.61
Underweight, n (%)	0 (0)
Normal, n (%)	33 (76.7)
Overweight, n (%)	10 (23.3)
Obesity, n (%)	0 (0)
Body fat (%), mean ± SD	15.2 ± 5.56
Waist perimeter (cm), mean ± SD	79.56 ± 7.25
Low, n (%)	42 (97.7)
High, n (%)	1 (2.3)
Very high, n (%)	0 (0)
Hip perimeter (cm), mean ± SD	94.91 ± 6.06
Waist–hip ratio, mean ± SD	0.84 ± 0.04
Arm circumference (cm), mean ± SD	31.87 ± 3.42
Thigh circumference (cm), mean ± SD	54.87 ± 4.68
Tricipital skinfold (mm), mean ± SD	11.95 ± 4.61
Nutritional variables	
HEI-S, mean ± SD	54.36 ± 10.45
Healthy, n (%)	8 (18.6)
Needs changes, n (%)	26 (60.5)
Unhealthy, n (%)	9 (20.9)
MEDAS-14, mean ± SD	4.65 ± 1.86
High adherence, n (%)	0 (0)
Moderate adherence, n (%)	13 (30.2)
Low adherence, n (%)	30 (69.8)
Physical variables	
ROM shoulder external rotation (°), mean ± SD	54.28 ± 8.75
ROM shoulder internal rotation (°), mean ± SD	98.23 ± 15.95
Hand grip strength (kg), mean ± SD	42.07 ± 9.65
Push strength—overall (kg), mean ± SD	74.75 ± 23.95
Push strength—rowing specific (kg), mean ± SD	71.86 ± 20.82
Functional variables	
Countermovement jump (cm), mean ± SD	29.5 ± 5.58
Push-up test (n), mean ± SD	38.02 ± 15.77
Sit-to-stand test (n), mean ± SD	55.47 ± 15.78
Sport-related variables	
Rowing experience (years)	5.77 ± 5.19
Training (hours/week)	8.99 ± 3.33
Shoulder injury during sports career (1 = yes)	6 (14%)
Back injury during sports career (1 = yes)	5 (11.6%)

SD: Standard deviation; BMI: body mass index; HEI-S: Healthy Eating Index-Spanish; MEDAS-14: Tool of Adherence to the Mediterranean Diet; ROM: range of motion; %: percentage; °: degrees; cm: centimetres; mm: millimetres; kg: kilograms.

**Table 2 ijerph-19-13518-t002:** Differences between seniors and U21 athletes.

Variable	Senior Group (n = 20) Mean ± SD	U21 Group (n = 23) Mean ± SD	Mean Difference (95% CI)
Exercise motivation			
Motivation to exercise—Physical	1.76 ± 0.12	1.82 ± 0.11	0.05 (−0.37 to 0.27)
Motivation to exercise—Social	1.34 ± 0.14	1.31 ± 0.13	0.02 (−0.37 to 0.42)
Motivation to exercise—Psychological	1.51 ± 0.11	1.56 ± 0.11	0.05 (−0.36 to 0.26)
Anthropometric variables			
BMI (kg/m^2^)	23.86 ± 2.31	22.75 ± 2.81	1.10 (−0.49 to 2.70)
Body fat (%)	15.00 ± 4.93	15.37 ± 6.16	−0.37 (−3.84 to 3.11)
Waist perimeter (cm)	81.07 ± 6.46	78.24 ± 7.77	2.83 (−1.61 to 7.27)
Hip perimeter (cm)	95.03 ± 6.24	94.80 ± 6.04	0.22 (−3.56 to 4.01)
Waist–hip ratio	0.85 ± 0.05	0.82 ± 0.04	0.03 ^a^ (0.01 to 0.05)
Arm circumference (cm)	33.06 ± 3.61	30.83 ± 2.94	2.24 ^a^ (0.21 to 4.25)
Thigh circumference (cm)	55.71 ± 4.24	54.13 ± 5.01	1.58 (−1.30 to 4.46)
Tricipital skinfold (mm)	11.30 ± 4.16	12.52 ± 4.98	−1.22 (−4.07 to 1.63)
Nutritional variables			
MEDAS-14	4.45 ± 1.98	4.82 ± 1.78	−0.38 (−1.53 to 0.78)
HEI-S	54.53 ± 9.14	54.21 ± 11.67	0.31 (−6.23 to 6.83)
Physical variables			
ROM shoulder external rotation (°)	53.85 ± 8.82	54.65 ± 8.86	−0.80 (−6.26 to 4.66)
ROM shoulder internal rotation (°)	95.75 ± 20.89	100.39 ± 9.90	−4.46 (−14.50 to 5.22)
Hand grip strength (kg)	47.70 ± 9.58	37.17 ± 6.69	10.53 ^b^ (5.49 to 15.46)
Push strength (kg)	89.87 ± 24.45	61.61 ± 13.83	28.26 ^b^ (16.24 to 40.30)
Push strength—rowing specific (kg)	84.30 ± 21.39	71.86 ± 20.82	23.57 ^b^ (12.51 to 34.01)
Functional variables			
Countermovement jump (cm)	31.26 ± 6.35	28.04 ± 4.49	3.21 (−0.17 to 6.60)
Push-up test (n)	47.15 ± 15.72	30.08 ± 10.96	17.06 ^b^ (17.06 to 8.80)
Sit-to-stand test (n)	58.68 ± 7.63	52.83 ± 8.27	5.86 ^a^ (0.85 to 10.86)
Sport-related variables			
Rowing experience (years)	9.25 ± 0.91	2.74 ± 0.85	7.41 ^b^ (7.42, 11.08)
Training (hours/week)	9.55 ± 0.74	8.5 ± 0.70	1.05 (−1.01, 3.11)
Shoulder injury during sports career (1 = yes)	0.25 ± 0.08	0.04 ± 0.07	0.21 ^a^ (0.00 to 0.42)
Back injury during sports career (1 = yes)	0.25 ± 0.07	0.00 ± 0.06	0.25 ^b^ (0.06 to 0.44)

Note: ^a^
*p* < 0.05; ^b^
*p* < 0.01; SD: standard deviation; CI: confidence interval; BMI: body mass index; HEI-S: Healthy Eating Index-Spanish; MEDAS-14: Tool of Adherence to the Mediterranean Diet; ROM: range of motion. %: percentage; °: degrees; cm: centimeters; mm: millimeters; kg: kilograms.

**Table 3 ijerph-19-13518-t003:** Multivariate regression analysis.

	1RM Rowing Strength—Test (n = 42)			
	B	SE	β	95% CI
Exercise motivation				
Motivation to exercise—Physical	5.64	1.82	0.27 ^b^	[1.86, 9.41]
Motivation to exercise—Social	−1.16	1.46	−0.06	[−4.18, 1.86]
Motivation to exercise—Psychological	2.11	1.63	0.10	[−1.47, 5.52]
Anthropometric variables				
Age (years)	−0.27	2.61	−0.01	[−5.69, 5.16]
Gender (1 = female)	18.77	8.15	0.34 ^c^	[1.83, 35.71]
Arm circumference	1.11	2.99	0.05	[−5.11, 7.33]
Thigh circumference	3.33	1.88	0.16 ^d^	[−0.58, 7.24]
Body fat	−7.26	2.43	−0.35 ^b^	[−12.32, −2.20]
Tricipital skinfold	−7.19	2.80	−0.35 ^b^	[−13.01, −1.38]
BMI	8.50	3.20	0.41 ^a^	[1.85, 15.15]
Waist–hip ratio ^e^	0.30	2.25	0.02	[−4.37, 4.98]
Nutritional variables				
MEDAS-14	−1.16	1.61	−0.06	[−4.51, 2.20]
HEI-S	−0.23	1.51	−0.01	[−3.37, 2.91]
Physical variables				
ROM ER	0.31	0.10	0.03	[−2.35, 3.65]
ROM IR	4.77	1.66	0.23 ^b^	[1.31, 8.22]
Hand grip strength	3.34	2.38	0.16	[−1.61, 8.29]
Push strength—overall	7.60	3.30	0.37	[0.73, 14.46]
Sport-related variables				
Rowing experience (years)	7.18	2.67	0.35 ^b^	[1.62, 12.74]
Training (hours/week)	−2.00	1.45	−0.10	[−5.02, 1.02]
Shoulder injury during sports career (1 = yes)	−14.44	4.80	−0.24 ^b^	[−24.42, −4.46]
Back injury during sports career (1 = yes)	−6.30	4.27	−0.10	[−15.16, 2.59]
	R^2^ =	0.94 ^a^	f^2^ = 14.62	1 − β = 1.00

Note: ^a^
*p* < 0.0001; ^b^
*p* < 0.01; ^c^
*p* < 0.05; ^d^
*p* < 0.10; 1 − β = achieved statistical power; f^2^ = Cohen’s effect size. All continuous variables were transformed to Z-scores to allow meaningful comparison among predictors. ^e^ Hip and waist perimeter were excluded from the model due to their collinearity with waist–hip ratio. One participant with missing values was excluded from the analyses.

## Data Availability

Data are available upon request.

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
