# Peer review of "Canoe polo Athletes’ Anthropometric, Physical, Nutritional, and Functional Characteristics and Performance in a Rowing Task: Cross-Sectional Study"

_ijerph, 2022, doi:10.3390/ijerph192013518_

Round 1
Reviewer 1 Report
This study aims: (1) to analyse the anthropometric, nutritional, physical, functional, and psychosocial characteristics of canoe polo athletes; (2) to explore the relative contribution of these variables in predicting performance in a rowing-specific strength exercise (as proxy of individual performance in canoe polo); (3) to evaluate the differences in these variables between senior and U21 athletes.
Personally, I found this study interesting because it could provide useful information to coaches, athletes and it could be very important from physical conditioning perspective. The findings could have useful practical implications, especially concerning predictive actions related to the specific performance in this discipline and the development of better training and rehabilitation programs. However, after reading and analysing it in more detail, I have some comments for improving even more the quality of this article.
I recommend rewriting the title, because health clinical outcomes do not seem to me to be related to the real objective of the manuscript. In addition to comparing between groups, the data have been taken to be predictors of sports performance through the 1RM rowing test, in short, I don't see any relation to health clinical outcomes.
In the abstract, the participants' information should appear before the type of statistical study conducted.
The introduction is too brief, without going in depth into the variables of study and their importance in this particular discipline.
Within the methodology, in the participants section, the first paragraph belongs rather to procedure, the second paragraph would be ok, but the description of the sample and its main characteristics (which are however included in results, which seems rather strange to me) would be completely missing.
Having stated this, the first part of the results should be in the methodology, specifically in the section on participants (Figure 1).
One question I would like to ask is about the last sentence of the results: “A second unpredicted, yet plausible finding was that prior shoulder injuries (β = -.24***) reduced the 1RM Rowing test score”, Could these results influence the performance prediction found among certain study variables for the 1RM rowing test? and, would it be appropriate to use this variable related to previous injuries as a possible covariate in a specific statistical analysis?
Discussion. Somewhat brief, as was already the case with the introduction. I think the authors should go into more depth in general terms, providing more contrast with other situations and giving a greater number of arguments to support the results obtained. Functional tests, for example, have not really been discussed, providing information that is not very relevant to the state of the question. It would be appropriate to include future research lines or possible investigations.
Please, revise the bibliographic references in accordance with the journal's regulations, as they do not include the “doi” for the articles referenced.
Thank you very much for your consideration in reviewing this manuscript.
I hope to be helpful again on future occasions.
Reviewer 2 Report
There are inconsistencies regarding the abbreviations - 1RM/ 1MR
For the triceps skinfold, please provide the measurement technique and reference.
Table 1 - How are „Shoulder Injury during sport career” and „ Back injury during sport career” reported - as the number of „Yes”? If so, I don`t think that reporting this variable as mean and SD is appropriate. Maybe number and percentage.
